# Single-cell and single-nucleus RNA-seq uncovers shared and distinct axes of variation in dorsal LGN neurons in mice, non-human primates, and humans

Trygve E Bakken[1†], Cindy TJ van Velthoven[1†], Vilas Menon[1,2†], Rebecca D Hodge[1], Zizhen Yao[1], Thuc Nghi Nguyen[1], Lucas T Graybuck[1], Gregory D Horwitz[3], Darren Bertagnolli[1], Jeff Goldy[1], Anna Marie Yanny[1], Emma Garren[1], Sheana Parry[1], Tamara Casper[1], Soraya I Shehata[1], Eliza R Barkan[1], Aaron Szafer[1], Boaz P Levi[1], Nick Dee[1], Kimberly A Smith[1], Susan M Sunkin[1], Amy Bernard[1], John Phillips[1], Michael J Hawrylycz[1], Christof Koch[1], Gabe J Murphy[1], Ed Lein[1], Hongkui Zeng[1], Bosiljka Tasic[1]*

[1]Allen Institute for Brain Science, Seattle, United States; [2]Department of Neurology, Columbia University Medical Center, New York, United States; [3]Washington National Primate Research Center and Department of Physiology and Biophysics, University of Washington, Seattle, United States

*For correspondence:
bosiljkat@alleninstitute.org

†These authors contributed equally to this work

Competing interests: The authors declare that no competing interests exist.

**Abstract** Abundant evidence supports the presence of at least three distinct types of thalamocortical (TC) neurons in the primate dorsal lateral geniculate nucleus (dLGN) of the thalamus, the brain region that conveys visual information from the retina to the primary visual cortex (V1). Different types of TC neurons in mice, humans, and macaques have distinct morphologies, distinct connectivity patterns, and convey different aspects of visual information to the cortex. To investigate the molecular underpinnings of these cell types, and how these relate to differences in dLGN between human, macaque, and mice, we profiled gene expression in single nuclei and cells using RNA-sequencing. These efforts identified four distinct types of TC neurons in the primate dLGN: magnocellular (M) neurons, parvocellular (P) neurons, and two types of koniocellular (K) neurons. Despite extensively documented morphological and physiological differences between M and P neurons, we identified few genes with significant differential expression between transcriptomic cell types corresponding to these two neuronal populations. Likewise, the dominant feature of TC neurons of the adult mouse dLGN is high transcriptomic similarity, with an axis of heterogeneity that aligns with core vs. shell portions of mouse dLGN. Together, these data show that transcriptomic differences between principal cell types in the mature mammalian dLGN are subtle relative to the observed differences in morphology and cortical projection targets. Finally, alignment of transcriptome profiles across species highlights expanded diversity of GABAergic neurons in primate versus mouse dLGN and homologous types of TC neurons in primates that are distinct from TC neurons in mouse.

## Introduction

The dorsal lateral geniculate nucleus (dLGN/LGd) of the thalamus receives visual information from the retina and projects to the primary visual cortex (V1/VISp) (*Jones, 2007*) via excitatory thalamo-cortical (TC) projection neurons. In addition, the dLGN receives modulatory inputs from diverse structures including V1, thalamic reticular nucleus, and brainstem nuclei that play an important role in visual processing and spatial awareness (*Saalmann and Kastner, 2011*; *Ling et al., 2015*;

*Okigawa et al., 2021*). dLGN also contains GABAergic interneurons, which tune visual responses by providing feedforward inhibition to TC neurons and have dual developmental origins (*Jager et al., 2021*) and variable electrophysiological properties (*Leist et al., 2016*).

In mammals with high visual acuity, including primates, carnivores, and some rodents (e.g., squirrel), dLGN is composed of three classes of TC neurons that are anatomically segregated into layers (*Sherman, 2020*). The three groups of TC neurons represent a well-documented example of distinct cell types that transmit parallel information streams in the central nervous system. In the non-human primate and cat, these cells differ dramatically in size, receive input from different types of retinal ganglion cells (RGCs), innervate different layers of V1, and respond preferentially to distinct features of visual stimuli (*Livingstone and Hubel, 1988*; *Callaway, 2005*). Magnocellular (M) TC neurons have large somata, receive input from rod photoreceptors, have larger receptive fields, provide higher contrast gain, and project to layer 4Cα of V1. In contrast, parvocellular (P) TC neurons receive input from cone photoreceptors, have smaller receptive fields, lower contrast gain, and project to layer 4Cβ of V1 (*Reid and Shapley, 2002*; *Jeffries et al., 2014*). The cell bodies of koniocellular (K) TC neurons reside in thin layers in between the M and P layers. At least some K neurons receive input from short-wavelength cones and project to layers 2–3 of V1 (*Hendry and Reid, 2000*). M, P, and K neurons express distinct marker genes in macaque, although M and P neurons are surprisingly similar based on bulk microarray profiling of dissected M and P layers (*Murray et al., 2008*).

Although macaque and human differ from mice in visual acuity and color vision, the structure of the visual system is broadly conserved across species (*Huberman and Niell, 2011*). Like in primates, RGCs provide convergent input to dLGN TC neurons in mice (*Peng et al., 2019*). The mouse dLGN is not clearly laminated but can be divided into two subregions – core and shell – that receive different retinal input and project to different layers of V1 (*Román Rosón et al., 2019*). Shell neurons preferentially receive input from direction-selective RGCs and project to layers 1–3 of V1, whereas core neurons mostly receive input from non-direction-selective RGCs and project to layer 4 (*Cruz-Martín et al., 2014*; *Seabrook et al., 2017*). In addition, based on their dendritic morphology, the dLGN neurons have been classified into X, Y, and W types (*Krahe et al., 2011*). The correspondence between projection targets and local dendritic morphology has not been established, and it is unknown whether transcriptomically distinct populations of TC neurons exist in mouse dLGN.

Recent advances in single-cell and single-nucleus RNA-sequencing (scRNA-seq and snRNA-seq) provide a powerful, complementary approach to anatomical studies to delineate and distinguish types of neurons on the basis of their genome-wide gene expression profiles (*Darmanis et al., 2015*; *Zeisel et al., 2015*; *Tasic et al., 2016*; *Saunders et al., 2018*; *Tasic et al., 2018*; *Zeisel et al., 2018*; *Hodge et al., 2019*; *Bakken et al., 2020*). These techniques can be used to examine the conserved and unique features of cell types in different species (*La Manno et al., 2016*; *Hodge et al., 2019*; *Bakken et al., 2020*; *Krienen et al., 2020*). For this study, we profiled single-nucleus (sn) or single-cell (sc) transcriptomes of dLGN of adult human, macaque, and mouse. For each species, we defined transcriptomic cell types and characterized their proportions, spatial distributions, and marker genes by sc/snRNA-seq. We also aligned RNA-seq data across species to determine the conservation of GABAergic interneuron types, correspondence of mouse core- and shell-enriched TC neurons to primate M, P, and K types, and primate specializations associated with their exceptional vision.

## Results

### Single-cell and -nucleus transcriptomic profiling

We used our previously described experimental approach (*Tasic et al., 2016*; *Bakken et al., 2018*; *Tasic et al., 2018*; *Hodge et al., 2019*; *Figure 1*, Materials and methods) to isolate and transcriptomically profile nuclei from macaque (*Macaca nemestrina* and *Macaca fascicularis*) and human dLGN, as well as cells from mouse dLGN. Nuclei were collected from microdissected anatomically defined regions: M or P layers of non-human primate dLGN; K, M, or P layers of human dLGN; and shell or core of mouse dLGN. We note that the microdissections are imperfect and likely include neighboring regions. This is especially evident for the K layers, which are very thin; their dissections inevitably included cells from neighboring M and P layers. Adjacent thalamic nuclei were also sampled, including ventral pulvinar from a single macaque donor, and lateral posterior (LP) and ventral

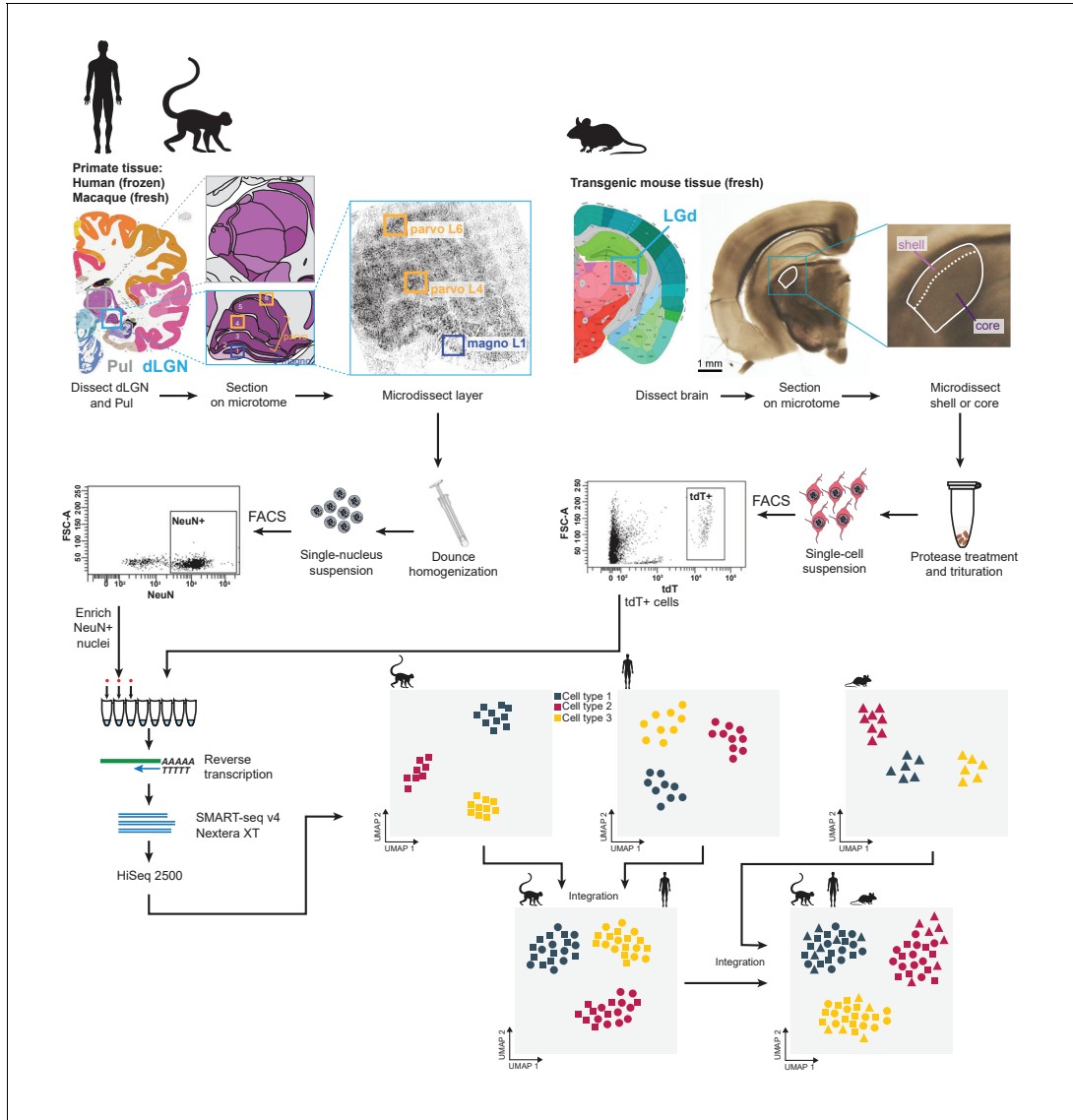

**Figure 1.** Experimental and data analysis workflow. Dorsal lateral geniculate nucleus (dLGN) was dissected from postmortem human brain and acutely collected macaque and mouse brain according to the Allen Brain Atlas. Each sample was used to obtain single-cell or -nucleus suspensions. Individual cells or nuclei were sorted into eight-well strip PCR tubes by FACS and lysed. SMART-Seq v4 was used to reverse-transcribe and amplify full-length cDNAs. cDNAs were then tagmented by Nextera XT, PCR-amplified, and processed for Illumina sequencing. Initial clustering of single-cell or single-nucleus transcriptomes was performed independently for human, macaque and mouse. To further distinguish M and P types, the human data werectustered with the macaque data. For cross-species comparison, macaque, human, and mouse data were co-clustered withSeurat v3.
The online version of this article includes the following figure supplement(s) for figure 1:

**Figure supplement 1.** Total reads and gene detection rates in the datasets.

lateral geniculate (LGv) nuclei from several mice. Single cells or nuclei were isolated by Fluorescence-Activated Cell Sorting (FACS) and enriched for neurons based on labeling with neuronal markers (NeuN in primates and tdTomato [tdT] in mouse). All single cells and nuclei were processed with SMART-seq v4 (Clontech) and Nextera XT (Illumina) and sequenced on HiSeq 2500 (Illumina). RNA-seq reads were aligned to corresponding genomes using the STAR aligner (*Dobin et al., 2013*). Gene expression was quantified as the sum of intronic and exonic reads per gene, normalized as counts per million (CPM), and $\log_2$-transformed as previously described (*Tasic et al., 2018*; *Hodge et al., 2019*). The Seurat v3 R package was used for clustering (*Butler et al., 2018*; *Stuart et al., 2019*) (Materials and methods). We report on 2003 macaque, 1209 human, and 2118

mouse QC-qualified single-cell and -nucleus transcriptomes with cluster-assigned identity (*Figure 1—figure supplement 1A*, *Supplementary file 1*). Samples were sequenced to a median depth of 1.3 million reads/nucleus for macaque, 2.4 million reads/nucleus for human, and 2.5 million reads/cell for mouse (*Figure 1—figure supplement 1B*). Median gene detection in macaque and human nuclei (~6000 and 6200, respectively) is lower than in mouse cells (~9000, respectively) (*Figure 1—figure supplement 1C*).

## Transcriptomic cell types in macaque dLGN and pulvinar

Single nuclei were isolated from three macaque donors across two species from dLGN and pulvinar. The nuclei were subjected to snRNA-seq, mapped to the *Macaca mulatta* genome, clustered with Seurat, and the relationship among clusters was explored in 2D-UMAP projections. This projection revealed heterogeneity within clusters that is driven by donor identity (*Figure 2—figure supplement 1A*). 1026–1438 genes were significantly differentially expressed (greater than twofold change, FDR < 0.01) between pairs of donors for glutamatergic neurons. Based on ontology enrichment analysis, these genes were associated with neuronal signaling and connectivity and not with metabolic or activity-dependent processes (*Supplementary files 2* and *3*).

To explore cell-type diversity shared across donors, we used the fastMNN implementation of Mutual Nearest Neighbors (MNN), which enables more accurate integration of imbalanced datasets compared to canonical correlation analysis (CCA) (*Haghverdi et al., 2018*; *Figure 2—figure supplement 1B*, Materials and methods). By removing donor-specific signatures, we defined nine shared neuronal types in macaque dLGN (*Figure 2A, B*, *Figure 2—figure supplement 1C*). We assigned cell-type identities based on known marker genes and dissection location (*Figure 2—figure supplement 1D*). The neuronal taxonomy has two major branches, GABAergic and glutamatergic (*Figure 2A*), that further branch into four and five types, respectively. We identified two distinct K types (Kap and Kp, *Figure 2A*) that express K-specific markers *CAMK2A* and *PRKCG* (*Murray et al., 2008*). The Kp-type selectively expresses *PENK* (*Supplementary file 4*). *PENK*-expressing cells are enriched in posterior K1 and K2 layers (*Figure 2—figure supplement 1E*).

The Pulv cluster corresponds to pulvinar TC neurons and expresses *GRIK3* and *LHX2* (*Figure 2A*, *Supplementary file 4*; *Jones and Rubenstein, 2004*). The M projection neurons express previously reported markers *ABHD17A* (*FAM108A*), *BRD4*, *CRYAB*, *EEF1A2*, *IL15RA*, *KCNA1*, *NEFM*, *PPP2R2C*, and *SFRP2* (*Murray et al., 2008*), and P projection neurons express *FOXP2* (*Iwai et al., 2013*) and *TCF7L2* (*Murray et al., 2008*). We also identified many novel M and P markers that are shared across donors (*Figure 2C*).

## Transcriptomic cell types in human dLGN

We define six neuronal and four non-neuronal types in human dLGN (*Figure 3A, B*, *Figure 3—figure supplement 1A–C*) by transcriptomically profiling individual nuclei isolated from three postmortem donors using the same methods as described above for macaque. The neuronal taxonomy has two major branches: GABAergic and glutamatergic, which further branch into three types each (*Figure 3A*). Based on the expression of glutamatergic markers and K-specific markers (*CALB1*, *CAMK2A,* and *PRKCG*, *Figure 3A*, *Supplementary file 4*), two K types, Kap and Kp, could be identified (*Hendry and Reid, 2000*; *Murray et al., 2008*). Similar to the K types identified in macaque, the *PENK*-expressing K type (Kp) is limited to posterior K1 and K2 layers (*Figure 3—figure supplement 1D*). The remaining glutamatergic nuclei belong to a single cluster to which we assign the MP projection neuron identity based on the following observations: (1) it is the most numerous glutamatergic type that expresses the known M/P marker gene *PVALB* (*Figure 3A*; *Yan et al., 1996*), (2) it does not express K markers *CALB1*, *CAMK2A,* and *PRKCG,* and (3) it contains cells derived from both M and P layer dissections (*Figure 3—figure supplement 2A*).

We detect donor-related heterogeneity in the human MP cluster (*Figure 3C*). This heterogeneity is unlikely due to sampling different subregions of dLGN across donors because we did not find any significantly differentially expressed genes between anterior and posterior dLGN dissected from individual donors (*Figure 3—figure supplement 2B*). 877–1324 genes were significantly differentially expressed genes between pairs of human donors and were associated with ribosomal processing rather than neuronal function (*Supplementary files 2* and *3*). 130 of these genes were reported as upregulated in postmortem versus acute neurosurgical tissue (*Hodge et al., 2019*). These results

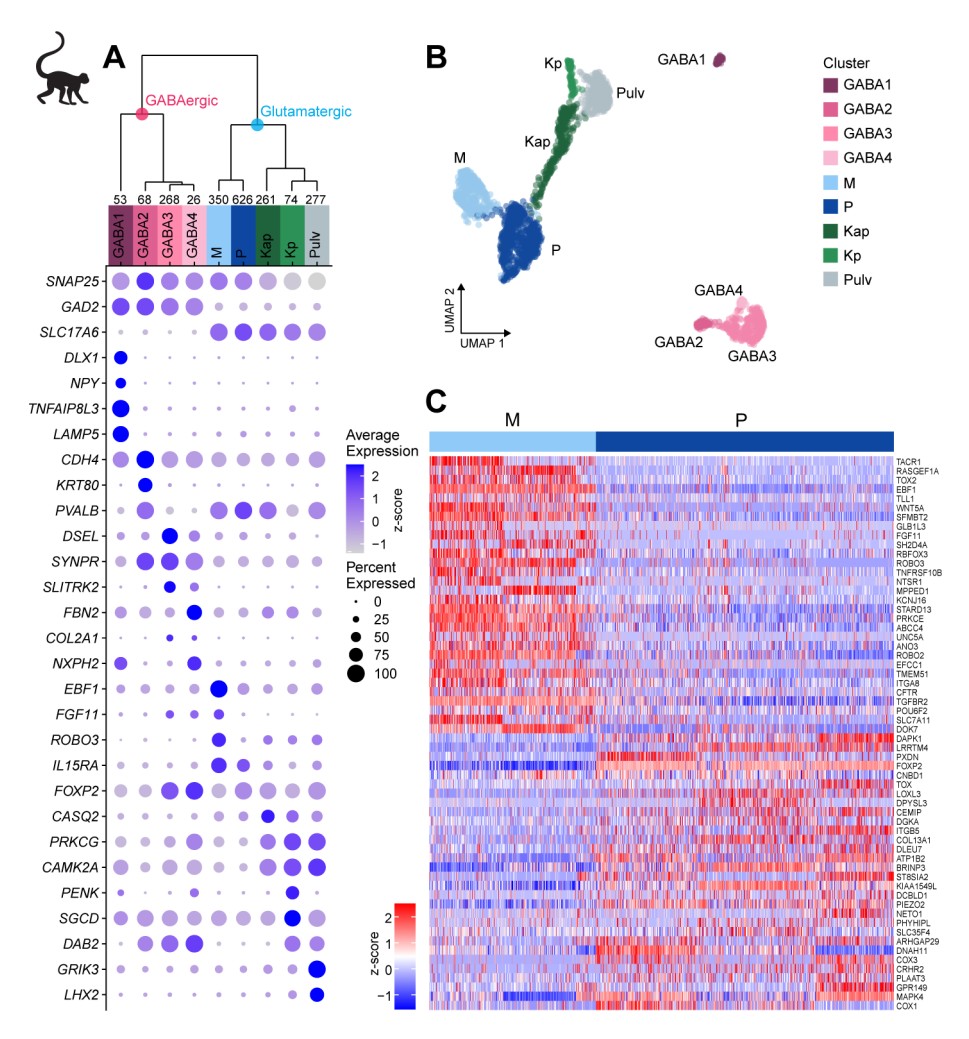

**Figure 2.** Neuronal cell-type taxonomy of macaque dorsal lateral geniculate nucleus (dLGN) and pulvinar by snRNA-seq. (**A**) Top: Neuronal cell-type taxonomy based on median cluster gene expression of 2000 differentially expressed genes in 2003 nuclei from three donors across two macaque species. Known marker genes and dissection location were used to assign molecular cluster identity. Numbers of nuclei in each cluster are indicated at the bottom of the dendrogram. Bottom: gene expression dot plot showing the relative expression of marker genes (y-axis) across all clusters (x-axis). The color intensity of the dots represents the average expression level, whereas the size of the dot represents the proportion of cells expressing the gene. (**B**) UMAP representation of macaque dLGN neurons colored by cluster. (**C**) Heatmap of RNA-seq expression z-scores computed for the top 60 differentially expressed genes expressed (p adj<0.05, log2(fold change) > 1) between the M and P clusters. Each column in the heatmap is an individual nucleus.

The online version of this article includes the following figure supplement(s) for figure 2:

**Figure supplement 1.** Donor-effect correction and cell type localization in macaque dorsal lateral geniculate nucleus (dLGN) cells.

suggest that dLGN tissue had variable quality across human donors, which limited discrimination of M and P cell populations. We were able to directly compare donor effects on gene expression in dLGN and cortex because the same donor brains were sampled in this study and our published study on cortical middle temporal gyrus (*Hodge et al., 2019*). Interestingly, only 456 genes were differentially expressed between donors for the most variable glutamatergic subclass, L2/3 IT neurons. This represents 40% of the differences seen in dLGN excitatory neurons for the same donors. Many factors could contribute to regional differences in donor effects, including sensitivity to tissue

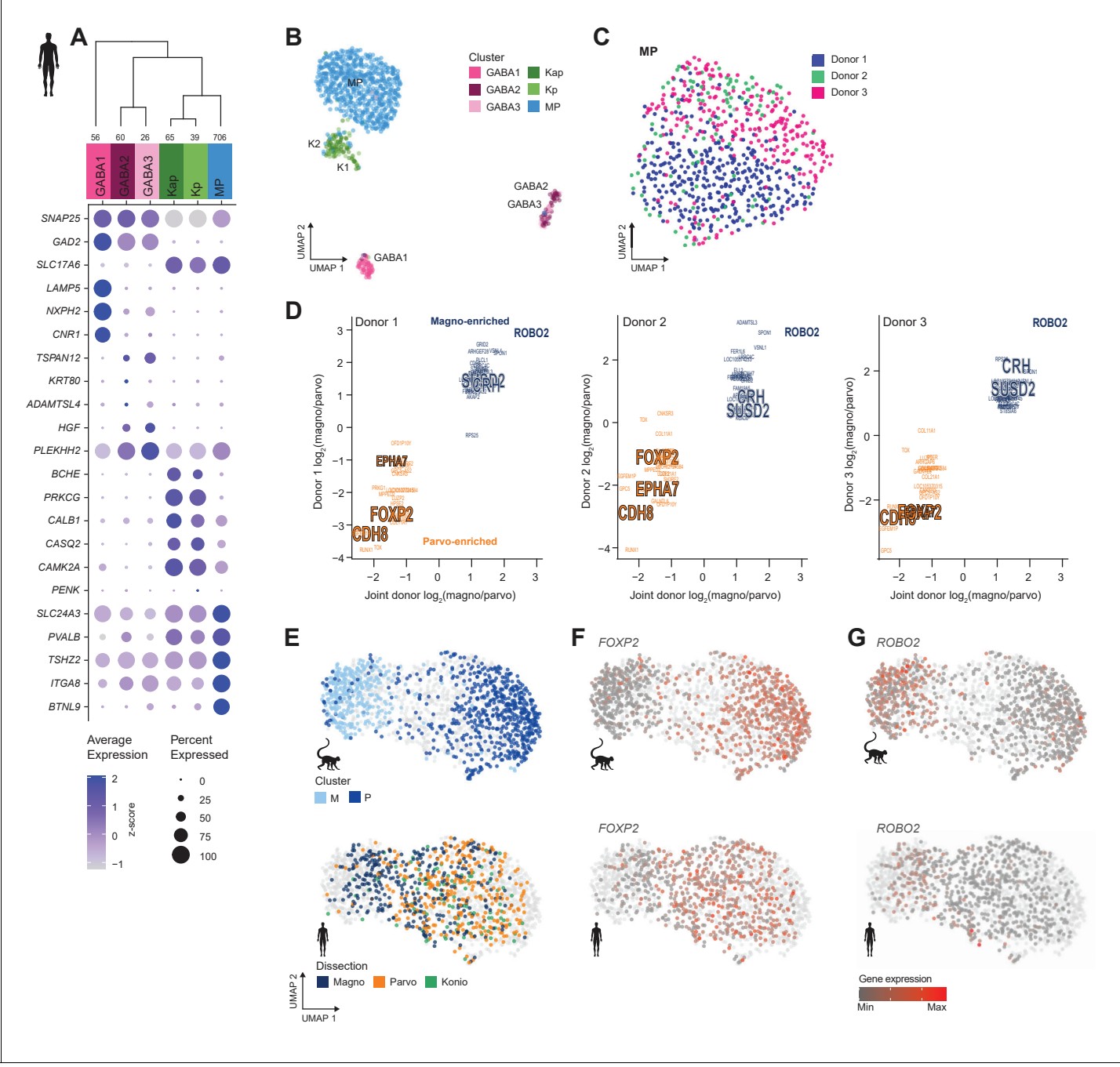

**Figure 3.** Neuronal cell-type taxonomy of human dorsal lateral geniculate nucleus (dLGN) by snRNA-seq. (A) Top: Cell-type taxonomy based on median cluster gene expression of 2000 differentially expressed genes and 946 nuclei from three donors. Known marker genes were used for cluster identity assignment. Numbers of nuclei in each cluster are indicated at the bottom of the dendrogram. Bottom: gene expression dot plot showing the relative expression of marker genes (y-axis) across all clusters (x-axis). (B) UMAP representation of human dLGN neurons colored by cluster. (C) UMAP representation of neurons within the MP cluster, colored by donor. (D) 32 single-nucleus marker genes show consistent enrichment between M and P dissections in three donors. Select marker genes that are differentially expressed (p adj<0.05, log2(fold change) > 1) are highlighted. (E) UMAP representation of joint analysis of human MP and macaque M and P nuclei using canonical correlation analysis (CCA). The macaque nuclei are labeled based on cluster identity defined by the macaque snRNA-seq clusteranalysis, whereas the human nuclei within the MP cluster are labeled by dissection ROI. (F, G) UMAP representation, as in panel (G), showing expression of the P marker *FOXP2* (F) or the M marker *ROBO2* in the macaque and human MP clusters (G).

The online version of this article includes the following figure supplement(s) for figure 3:

**Figure supplement 1.** Neuronal and non-neuronal cell types of human dLGN and marker gene expressionconfirmation.

*Figure 3 continued on next page*

*Figure 3 continued*

**Figure supplement 2.** Marker gene expression in M and P neurons in human dorsal lateral geniculate nucleus (dLGN).

processing, differential heterogeneity in cell states, or cell type-dependent effects of genetic background and environment.

Next, we looked for signatures of M and P types within individual donors. Among the MP nuclei from one donor, we observe a continuum of nuclei between M and P dissections (*Figure 3—figure supplement 2C*). To examine this continuum in more detail, we analyzed co-clustering matrices, which define how often nuclei are placed in the same cluster after 100 iterations of clustering. In all three donors, these co-clustering matrices show substructure within the MP type that corresponds to M and P dissections (*Figure 3—figure supplement 2B*). Examination of MP cells within each donor and comparison to joint donor signatures shows that many genes are consistently differentially expressed between M and P neurons (*Figure 3D*). We also confirmed expression of select marker genes in appropriate layers by RNA in situ hybridization (ISH) (*Figure 3—figure supplement 2E, F*), including newly discovered markers for M (*CRH* and *SUSD2*), P (*EPHA7*), or M and P (*BTLN9*) neurons.

The gene expression differences between glutamatergic neurons from M and P dissections in some of the human donors were subtle, but they were clearly observable in the macaque dataset (*Figure 2D*, *Figure 3—figure supplement 2G*). We therefore wondered if the MP continuum in human would align with the M and P cluster division in macaque. To investigate this possibility, we examined human nuclei and batch-corrected macaque nuclei together by employing CCA in Seurat v3. After the cross-species integration, the macaque M and P clusters remained well segregated, whereas the nuclei from the human MP cluster still formed a continuum. Encouragingly, the human MP continuum aligned with the macaque M and P clusters; that is, the human M-pole nuclei aligned with the macaque M cluster, and likewise, the human P-pole nuclei aligned with the macaque P cluster (*Figure 3E*). This alignment indicates that there is a shared signature between species in the M/P cell populations. *FOXP2* is a robust marker of P neurons (*Iwai et al., 2013*) and is enriched in the macaque P cluster and overlapping nuclei in the human MP cluster (*Figure 3F*) compared to other nuclei. Likewise, *ROBO2* is enriched in the macaque M cluster and overlapping human nuclei (*Figure 3G*). Despite the differences among donors, species, and nuclear quality, integration of transcriptomic data enabled us to identify a common axis of gene expression variation that is conserved among macaque and human and aligns with the previously defined M/P anatomical axis.

## Transcriptomic cell types in mouse dLGN, LP, and LGv

To profile cells from mouse dLGN while examining the reported diversity in TC neurons (*Krahe et al., 2011*; *Cruz-Martín et al., 2014*), we performed dissections that enriched for core and shell regions of dLGN. As noted before for macaque and human tissue, the microdissections are imperfect and should be considered enrichments for dissected regions of interest. We identified 12 neuronal and 3 non-neuronal types from mouse dLGN dissections (*Figure 4A, B*, *Figure 4—figure supplement 1A–C*). The 12 neuronal types could be further divided into 9 GABAergic and 3 glutamatergic types. However, due to the small size of dLGN in mice, we suspected that some of these types likely originated from neighboring areas that could not be clearly separated by microdissections. Therefore, as controls, we also profiled single cells from nearby thalamic nuclei, LP, and LGv. To assign anatomical location to clusters, we identified differentially expressed genes selective to each cluster and then examined expression patterns of these marker genes in the Allen Brain Atlas RNA ISH data (*Lein et al., 2007*; *Supplementary file 4*). Based on these marker genes, we assigned three GABAergic types to dLGN, whereas the remaining GABAergic types were likely from adjacent thalamic nuclei, including LP, LGv, and the reticular thalamic nucleus (RT, *Figure 4C*). Based on the dissection area as well as the differentially expressed genes, all glutamatergic cells from dLGN belong to a single cluster 'dLGN' (*Figure 4A–C*, *Figure 4—figure supplement 1D*). Cells within this cluster are not homogeneous, with the major axis of gene expression variation corresponding to the core vs. shell anatomical axis (*Figure 4D*, *Supplementary file 5*). These expression differences do not appear to be activity-dependent since there was minimal overlap with genes recently reported to be up- or downregulated in response to visual stimuli in mouse dLGN relay neurons

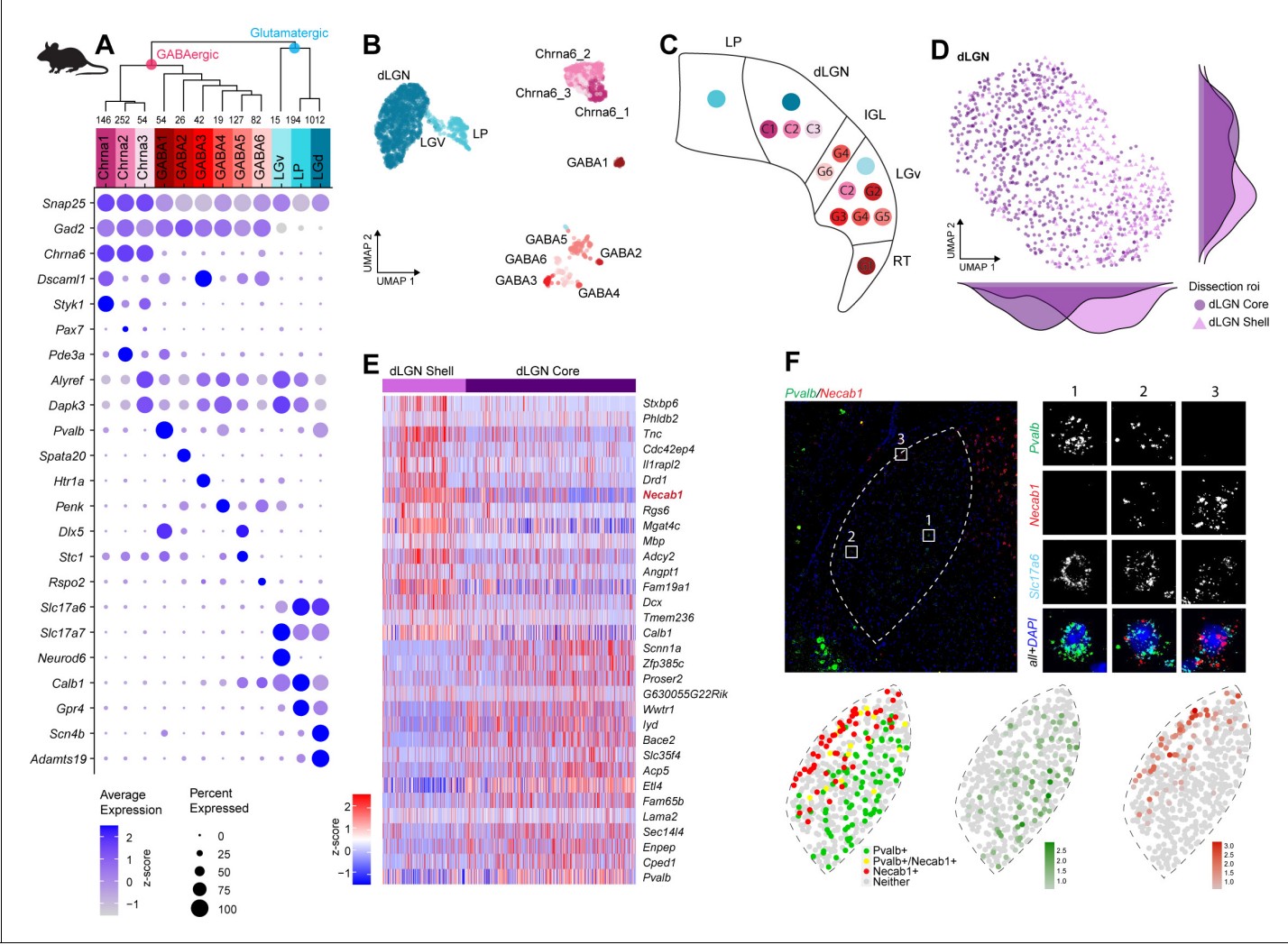

**Figure 4.** Neuronal cell-type taxonomy of mouse dorsal lateral geniculate nucleus (dLGN) and nearby regions by scRNA-seq. (A) Top: hierarchical taxonomy based on median cluster gene expression of >2000 differentially expressed genes and 2020 cells. Known and newly discovered marker genes were used to assign molecular cluster identity. Bottom: gene expression dot plot showing the relative expression of marker genes (y-axis) across all clusters (x-axis). (B) UMAP representation of mouse dLGN neurons colored by cluster. (C) Schematic representation of the relevant thalamic nuclei in the mouse brain with colored dots representing cell types identified in this study. Based on cell-type-specific marker expression and using the Allen Brain Atlas in situ hybridization (ISH) data, the anatomical location of cell types could be determined. (D) UMAP representation of neurons from the dLGN cluster colored by dissection ROI. The density plot in the margin shows the distribution of cells dissected from mouse dLGN-core (dark purple) and mouse dLGN-shell (light purple) along the x- and y-axes. (E) Heatmap of RNA-seq expression z-scores computed for the top 30 differentially expressed genes expressed (p adj<0.05, log2(fold change) > 1) between cells obtained from dLGN shell and core dissections belonging to dLGN cluster. The gene highlighted in red is confirmed by ISH in panel F. Each column in the heatmap is an individual sample. (F) Confirmation of differential expression of *Pvalb* and *Necab1* between shell and core of mouse dLGN by single-molecule fluorescence ISH by RNAscope.

The online version of this article includes the following figure supplement(s) for figure 4:

**Figure supplement 1.** Marker gene expression in mouse dorsal lateral geniculate nucleus (dLGN).

(*Cheadle et al., 2018*). To confirm this anatomical and gene expression heterogeneity in situ, we identified differentially expressed genes between cells isolated from shell and core of dLGN (*Figure 4E*) and validated expression of a subset of genes by multiplexed RNA ISH (*Figure 4F*, *Figure 4—figure supplement 1E, F*). We confirm our scRNA-seq findings that neurons in the shell express higher levels of *Necab1* and *Calb1*, and neurons in the core more highly express *Pvalb* and *Scn1a*. A small subset of neurons co-express *Pvalb* and *Necab1*, suggesting that there are 'intermediate' cells in dLGN that share shell- and core-like properties. In agreement with this finding, Calb1

protein has been previously reported to be more highly expressed in the dLGN shell compared to the core as measured by immunohistochemical labeling (*Grubb and Thompson, 2004*).

## Cross-species analysis of neuronal cell types in dLGN

To examine cross-species correspondence of transcriptomic cell types, we integrated the macaque and human snRNA-seq datasets with the mouse scRNA-seq dataset using CCA in Seurat v3 (*Figure 5A–C*) for a dataset of n = 4979 neurons. We included data from all thalamic nuclei to assess similarities of cell populations across regions and species. 2D-UMAP projections show extensive intermingling of GABAergic types across the three species and more separation between glutamatergic types (*Figure 5B, C*). To assess the correspondence, we generated an integrated taxonomy of the cell types (n = 17) identified by analysis of each species independently and compared this to the integrated clustering result (n = 10 cell types, *Figure 5D*). The integrated taxonomy has two major

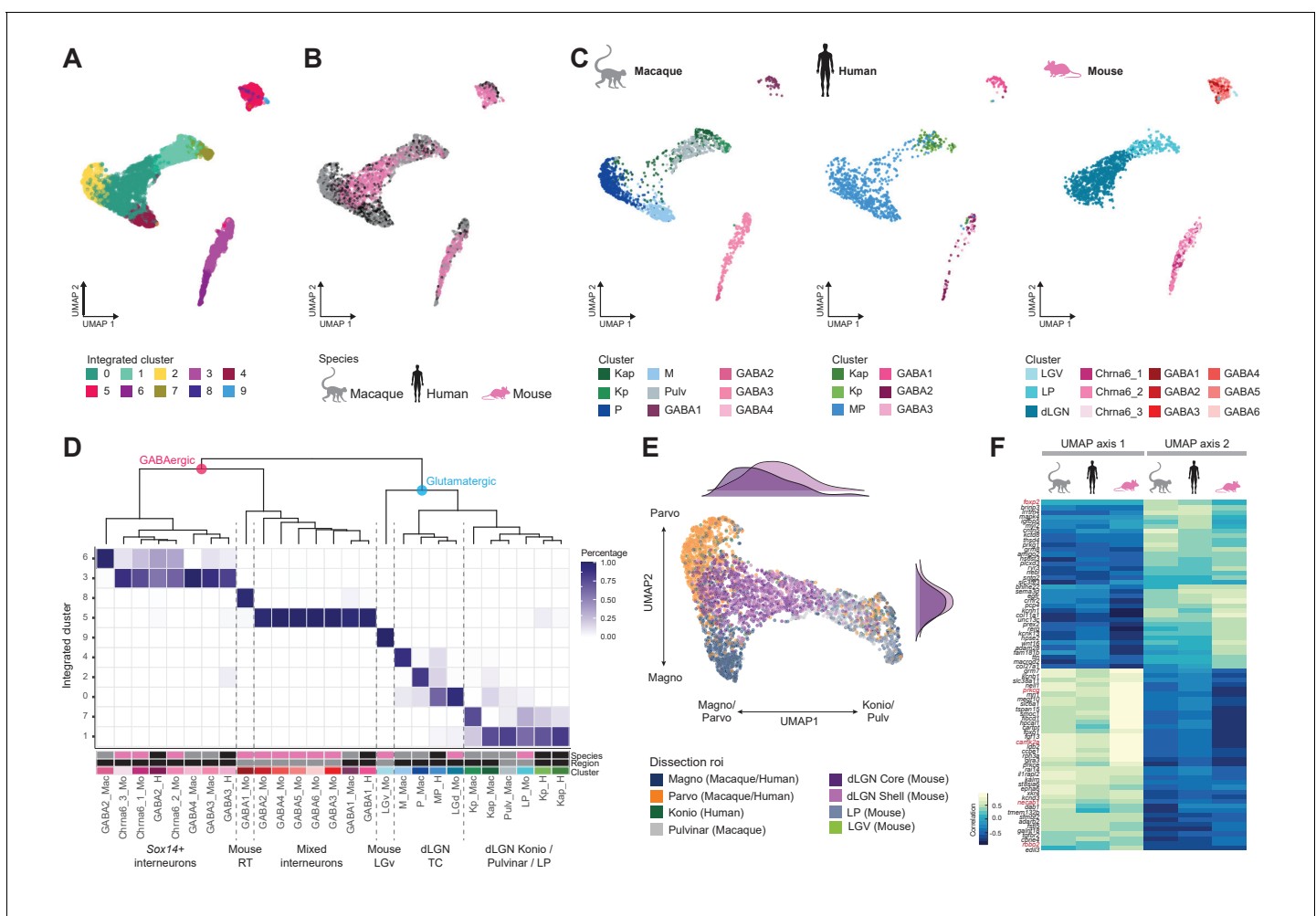

**Figure 5.** Cross-species integrative analysis. (**A–C**) All 4978 neurons from macaque, human, and mouse were integrated using Seurat v3. UMAP representation of the integrative analysis colored by integrated cluster call (**A**), colored by species (**B**), or colored by species-independent cluster call (**C**). (**D**) Correspondence of species-specific clustering and integrative clustering. The heatmap illustrates the proportion of the species-specific cell types contributing to the integrated cluster. The region color bar indicates the location of cell types in dorsal lateral geniculate nucleus (dLGN) (black) or in adjacent thalamic nuclei (gray). (**E**) Representation of glutamatergic neurons selected from UMAP as represented in panels (**A**) and (**B**) colored by dissection ROI. The density plot in the margin shows the distribution of cells dissected from mouse dLGN-core (dark purple) and mouse dLGN-shell (light purple) along the x- and y-axes. (**F**) Heatmap showing the Spearman correlation coefficient of the gene expression along the UMAP axes represented in panel (**C**) per species. For each axis, the top 30 genes are shown.

The online version of this article includes the following figure supplement(s) for figure 5:

**Figure supplement 1.** Cross-species comparison of GABAergic types.

branches: GABAergic and glutamatergic. The integrated GABAergic class contains 16 species-specific types that map to 4 integrated types (*Figure 5D*, *Figure 5—figure supplement 1A*). Cluster 8 from the integrated taxonomy contains only cells from the mouse GABA1 type. This type represents highly distinct RT neurons (*Figure 4B, C*) that were not sampled in primates and were profiled in mice likely due to imperfect dissections. The mouse GABAergic types can be roughly divided into two groups based on the expression of key transcriptional regulators (*Sox14*, *Lef1*, *Otx2*, *Nkx2-2*, and *Dlx5*) that reflect their developmental origin in the midbrain (*Sox14+*) or forebrain (*Sox14-*) (*Scholpp and Lumsden, 2010*; *Jager et al., 2021*). All mouse GABAergic types in dLGN and some types in the ventrolateral geniculate nucleus (LGv) and the intergeniculate leaflet (IGL) express *Sox14*, which is consistent with previous reports (*Sellers et al., 2014*; *Jager et al., 2016*). Similarly, two macaque and two human GABAergic types express *SOX14* and are homologous to the *Chrna6/Sox14*-expressing GABAergic dLGN types in mouse and form a distinct branch in the integrated taxonomy (*Figure 5D*). A second GABAergic branch includes the macaque GABA1 type, human GABA1 type, and mouse GABA2-6 types that are defined by expression of *Npy*, *Nkx2-2*, and *Dlx1/2/5/6* (*Figure 5—figure supplement 1B–D*). Interestingly, the mouse GABA2-6 types are not found in dLGN but are located only in the adjacent thalamic nuclei; IGL, LGv, and LP (*Figure 4C*). In human, the forebrain-derived GABA1 type was validated to be localized to dLGN based on RNA ISH (*Figure 5—figure supplement 1E*) and represents ~40% of GABAergic neurons. In contrast, in the macaque, the forebrain-derived GABA1 type represents only ~15% of GABAergic neurons in dLGN (*Figure 2A*, *Figure 5—figure supplement 1F*). In summary, we identify two major groups of GABAergic neurons that have distinct embryonic origins and are conserved across species with different proportions in dLGN.

Glutamatergic types did not align as clearly across species as GABAergic types, and mouse types were particularly distinct (*Figure 5C, D*). Homologies between macaque and human types, including K subtypes, were better resolved when mouse cells were excluded (*Figure 5—figure supplement 1G*). As expected, glutamatergic mouse LGv neurons clustered separately because this region was not sampled in primates. Transcriptomic signatures of mouse LP neurons resembled macaque inferior pulvinar neurons, located in what is considered the homologous structure in primates (*Harting et al., 1972*; *Baldwin et al., 2017*), as well as closely related K neurons. Like the integration of human and macaque data shown in *Figure 3F*, the human MP cells form a gradient along one axis with the macaque M and P types populating the distinct ends of that same gradient (*Figure 5C–E*). However, the mouse dLGN cluster variation not only aligns with this gradient of macaque and human M/P types, but rather spans both the continuum between the M and P types along UMAP axis 2 and the continuum between M/P and K/pulvinar/LP types on UMAP axis 1 (*Figure 5E*). Intriguingly, more shell- than core-dissected neurons from mouse dLGN resembled K/pulvinar neurons, consistent with reported similarities in their connectivity (*Bickford et al., 2015*). For each species, we correlated gene expression with position along these axes and found many genes with graded expression changes along both axes. There is clear conservation in the expression pattern of genes like *ROBO2*, *FOXP2*, and *CAMK2A* along UMAP axis 1, corresponding to the M/P to K/pulvinar difference (*Figure 5F*). These data show that despite conservation of cell types in the mature dLGN across species, there exist prominent differences in gene expression and cell-type proportions.

## Discussion

We used unsupervised clustering to define transcriptomic cell types in a well-studied part of the mammalian thalamus, dLGN, for human, macaque, and mouse. We examined the correspondence of excitatory TC neurons to previously described morphological, connectional, and physiological differences (*Hendry and Reid, 2000*; *Krahe et al., 2011*; *Cruz-Martín et al., 2014*) and identified cell-type homologies across species.

We find that primate K neurons are transcriptomically clearly distinguishable from their M and P counterparts, consistent with previous results (reviewed in *Hendry and Reid, 2000*). Moreover, we identify two transcriptomic types of K neurons that may correspond to two populations reported in macaque that have distinct laminar distributions in dLGN and projection patterns in V1 (*Casagrande et al., 2007*). We also find that K neurons are more transcriptomically similar to inferior pulvinar neurons than to M or P neurons. This is consistent with their shared inputs from retina and

superior colliculus and cortical projection targets (*Huo et al., 2019*) and supports a close functional relationship between K neurons and inferior pulvinar. M and P neurons are transcriptomically similar to each other in both macaque and human.

In contrast to primate, TC neurons in mouse dLGN cannot be grouped into discrete types based on gene expression, although many genes show graded expression differences between core and shell regions. Similar graded expression heterogeneity has been reported within and across many first- and higher-order thalamic nuclei (*Phillips et al., 2019*). Alignment of primate and mouse TC neurons revealed cell-type homologies between pulvinar and LP neurons, as expected. Mouse shell neurons aligned more with primate K neurons and core neurons aligned with M and P neurons, and some marker genes were conserved across species. Genes for calcium binding proteins were expressed in opposing gradients in TC neurons across mouse dLGN: *Pvalb*-expressing neurons were enriched in the core (and primate M and P neurons), *Calb1*- and *Necab1*-expressing neurons were enriched in the shell (and K neurons).

Our data revealed two homologous subclasses of GABAergic interneurons across species: *Sox14*-positive interneurons were present in dLGN in all species, and *Sox14*-negative interneurons were present in dLGN in human and macaque and only in thalamic nuclei adjacent to dLGN in mouse (*Figures 4C* and *5D*). Thus, primate dLGN exhibits increased diversity of GABAergic interneurons compared to mouse, potentially contributing to enhanced visual information processing required for more complex visually guided behaviors in primates. These two interneuron subclasses have consistent developmental origins across primates and rodents; *Sox14*-positive GABAergic neurons originate from midbrain, and *Sox14*-negative GABAergic neurons originate from forebrain (*Golding et al., 2014*; *Jager et al., 2021*). Interestingly, *Sox14*-negative, *Dlx1/2*-positive interneurons are progressively more common in larger-brained primates, representing less than 10% of GABAergic cells in marmoset dLGN (*Jager et al., 2021*), 15% in macaque, and 40% in human. This dramatic expansion in human may also be driven by human-specific migration of these interneurons from the ganglionic eminences that has not been observed in non-human primates, rodents, or other mammals (*Letinic and Rakic, 2001*).

Strong donor-specific molecular signatures were found in macaque and human datasets and could have several causes. Donor effects may be driven by the greater genetic diversity of primate donors than inbred mice. Alternatively, there may be donor differences in cell state, although differentially expressed genes were not enriched for previously identified activity-dependent genes (*Supplementary file 2*). Donor effects are unlikely due to profiling single nuclei versus single cells because we have previously shown excellent correspondence between cell-type taxonomies derived from cortical cells and nuclei (*Bakken et al., 2018*). M and P neurons could be more easily distinguished in macaque than human likely due to the use of acute surgical tissue in macaque versus frozen tissue with an 18.5–25 hr postmortem interval in human. Interestingly, primate donor effects were more prominent among TC neurons than GABAergic neurons in dLGN or human neocortex (*Hodge et al., 2019*), suggesting that cell types have different biological variability or robustness to technical artifacts.

In adult mouse dLGN, three distinct neuronal types, X-, Y-, and W-like, can be identified based on their morphology but have similar electrophysiological properties (*Krahe et al., 2011*; *Bickford et al., 2015*). Based on the unsupervised clustering presented in this article, we did not identify distinct transcriptomic types that correspond to these previously described morphological cell types. Likewise, in macaque and human, M and P neurons have highly distinct morphologies, connections, and locations in the dLGN, yet are transcriptomically similar. Other groups have made similar observations where cell types can be clearly distinguished based only on some properties (*Cadwell et al., 2016*; *Fuzik et al., 2016*; *Mayer et al., 2019*). In two cases in the mouse cortex, neurons located within the same cortical area but with different cortico-cortical projection patterns display relatively subtle differences in transcriptomic profiles (*Kim et al., 2020*; *Whitesell et al., 2021*). One explanation for this apparent discrepancy is that genes that shape the morphology and connectivity of TC neurons are expressed transiently during development. Indeed, a recent single-cell RNA-seq study of postnatal development of mouse dLGN showed increased transcriptional heterogeneity between postnatal day (P)10 and P16 compared to adult (*Kalish et al., 2018*). These two timepoints flank the onset of visual experience at time of eye-opening, which typically occurs between P12 and P14, indicating that transcriptional heterogeneity may peak during synaptogenesis and synaptic partner matching (*Hooks and Chen, 2006*; *Iwai et al., 2013*). A similar phenomenon

has been described in *Drosophila* where the transcriptomes of closely related types of projection neurons differ the most during circuit assembly and are highly similar in adult stage (*Li et al., 2017*; *Kurmangaliyev et al., 2020*; *Özel et al., 2021*). We speculate that maturing TC neurons during dLGN development may show more discrete transcriptomic signatures than we observe in the adult.

The discrepancy between the clear morphological, electrophysiological, and positional distinctions, on the one hand, and the more nuanced transcriptional differences among cell types, on the other hand, reinforces the notion that cell-type identification is best addressed by taking a multi-modal approach. For example, heterogeneity among mouse cerebellar molecular layer interneurons could only be clarified by joint characterization of gene expression, morphology, and physiological properties (*Kozareva et al., 2020*). This study reported continuous variation in gene expression in the unipolar brush cells that corresponded to distinct electrophysiological but not morphological properties. Multi-modal methods, such as Patch-seq (*Cadwell et al., 2016*; *Fuzik et al., 2016*), can define cell types based on morphology, connectivity, firing patterns, and other attributes relevant to their function in neural circuits (*Gouwens et al., 2020*; *Peng et al., 2020*; *Scala et al., 2020*). In the future, a clearer definition of discrete and continuous heterogeneity in transcriptomic landscapes may be enabled by new and improved experimental and analytical methods, such as spatial transcriptomics, that can comprehensively sample RNA transcripts from cells in situ (*Lein et al., 2017*; *Close et al., 2021*) and that could be performed after measurements of other neuronal properties such as morphology or in vivo activity.

# Materials and methods

## Key resources table

| Reagent type (species) or resource | Designation | Source or reference | Identifiers | Additional information |
|---|---|---|---|---|
| Strain, strain background (*Mus musculus*) | Mouse: B6.Cg-Gt (ROSA)$^{26Sortm14(CAG-tdTomato)Hze}$/J, Ai14(RCL-tdT) | The Jackson Laboratory | RRID:IMSR_JAX:007914 | |
| Strain, strain background (*Mus musculus*) | Mouse: B6J. Cg-*Gad2*$^{tm2(cre)Zjh}$/ MwarJ, Gad2-IRES-Cre | The Jackson Laboratory | RRID:IMSR_JAX:028867 | |
| Strain, strain background (*Mus musculus*) | Mouse: B6J.129S6 (FVB)-*Slc17a6*$^{tm2(cre)Lowl}$/ MwarJ, Slc17a6-IRES-Cre | The Jackson Laboratory | RRID:IMSR_JAX:028863 | |
| Strain, strain background (*Mus musculus*) | Mouse: B6J.129S6 (FVB)-*Slc32a1*$^{tm2(cre)Lowl}$/ MwarJ, Slc32a1-IRES-Cre | The Jackson Laboratory | RRID:IMSR_JAX:028862 | |
| Strain, strain background (*Mus musculus*) | Mouse: B6; 129S-*Snap25*$^{tm2.1(cre)Hze}$/ J, Snap25-IRES2-Cre | The Jackson Laboratory | RRID:IMSR_JAX:023525 | |
| Commercial assay or kit | SMART-Seq v4 Ultra Low Input RNA Kit for Sequencing | Takara | 634894 | |
| Commercial assay or kit | RNAscope Multiplex Fluorescent V2 Assay | Advanced Cell Diagnostics | 323100 | |
| Commercial assay or kit | RNAscope 2.5 HD Duplex Assay Kit | Advanced Cell Diagnostics | 322435 | |
| Software, algorithm | STAR 2.5.3 | PMID:23104886 | RRID:SCR_004463 | |
| Software, algorithm | Seurat | PMID:29608179 | PMID:29608179 | |
| Software, algorithm | fastMNN | PMID:29608177 | RRID:SCR_017351 | |
| Software, algorithm | ToppGene | PMID:19465376 | RRID:SCR_005726 | |

## Overall procedures and data analysis

Full experimental and data processing procedures are available at the Allen Institute web site within a detailed white paper: http://help.brain-map.org/display/celltypes/Documentation?preview=/8323525/10813526/CellTypes_Transcriptomics_Overview.pdf. Below, we list specific aspects that pertain only to this study.

## Mouse breeding and husbandry

All procedures were carried out in accordance with the Institutional Animal Care and Use Committee protocols 1508, 1510, and 1511 at the Allen Institute for Brain Science. Animals were provided food and water ad libitum and were maintained on a regular 12 hr day/night cycle at no more than five adult animals per cage. Animals were maintained on the C57BL/6J background. Experimental animals were heterozygous for the recombinase transgenes and the reporter transgenes. We utilized four Cre lines crossed to the tdT-expressing Cre reporter *Ai14* (*Madisen et al., 2010*): one pan-neuronal (*Snap25-IRES2-Cre*) (*Harris et al., 2014*), one pan-glutamatergic (*Slc17a6-IRES2-Cre*) (*Vong et al., 2011*), and two pan-GABAergic lines (*Gad2-IRES-Cre* and *Slc32a1-IRES-Cre*) (*Tong et al., 2008*; *Taniguchi et al., 2011*). Tissues were dissected from eight different donors. To dissect core and shell regions from mouse dLGN, block-face images were captured during slicing at 250 μm intervals. Slices were transferred into dissection dishes containing chilled, oxygenated ACSF. Brightfield and fluorescent images of each slice before and after ROI dissection were taken from the dissecting scope. To guide anatomical targeting of core- and/or shell-enriched dissections, boundaries were identified by trained anatomists, comparing the block-face image and the slice image to a matched plane of the Allen Mouse Brain Common Coordinate Framework version 3 (CCFv3) ontology *Wang et al., 2020*.

## Macaque tissue

The brain tissues of two adult *M. nemestrina* (southern pig-tailed macaque) and one *M. fascicularis* (crab-eating macaque) were obtained through the Tissue Distribution Program of the Washington National Primate Research Center and conformed to the guidelines provided by the US National Institutes of Health. All procedures were approved by the Institutional Animal Care and Use Committee of the University of Washington under protocol number 4277-01.

## Human tissue

Postmortem adult human brain tissue from three donors was collected after obtaining permission from decedent next-of kin. Postmortem tissue collection was performed in accordance with the provisions of the United States Uniform Anatomical Gift Act of 2006 described in the California Health and Safety Code section 7150 (effective 1/1/2008) and other applicable state and federal laws and regulations. The Western Institutional Review Board reviewed tissue collection processes and determined that they did not constitute human subjects research requiring institutional review board (IRB) review. In general, 3–5 slices were sufficient to capture the targeted region of interest, allowing for expression analysis along the anterior/posterior axis.

## Single-cell-/nucleus processing for sc/snRNA-seq

We used previously described procedures to perform single-cell and single-nucleus RNA-seq (*Bakken et al., 2018*; *Tasic et al., 2018*). In brief, cells and nuclei were isolated by FACS: macaque and human nuclei were stained with the neuronal marker NeuN and NeuN+ nuclei were sorted, whereas mouse cells were collected from several transgenic Cre-driver lines that preferentially label neuronal cells. We reverse-transcribed mRNA and amplified cDNA using Smart-seq V4 (Clontech), prepared sequencing libraries using Nextera XT (Illumina), and sequenced the libraries using HiSeq2500 (Illumina). We employed previously described quality control (QC) steps (*Bakken et al., 2018*; *Tasic et al., 2018*) to arrive to the final datasets (*Figure 1—figure supplement 1*).

## Data processing and analysis

Processing of sequencing data was performed as described before (*Bakken et al., 2018*; *Tasic et al., 2018*; *Hodge et al., 2019*; *Bakken et al., 2020*). For mouse, raw read (fastq) files were aligned to the mm10 mouse genome sequence (*Church et al., 2011*) with the RefSeq transcriptome

version GRCm38.p3 (current as of 01/15/2016) and updated by removing duplicate Entrez gene entries from the gtf reference file. For human, raw read files were aligned to the GRCh38 human genome sequence (Genome Reference Consortium, 2011) with the RefSeq transcriptome version GRCh38.p2 (current as of 4/13/2015) and likewise updated by removing duplicate Entrez gene entries from the gtf reference file. For analysis of transcriptomes of *M. nemestrina* (southern pig-tailed macaque) and *M. fascicularis* (crab-eating macaque), which are the species of macaque used for experiments, we used the genome assembly and annotation Mmul_10 of *M. mulatta* (rhesus macaque). Alignment to the genome was performed using STAR v2.5.3 (*Dobin et al., 2013*). Only uniquely aligned reads were used for gene quantification. The observed bimodal pattern in reads per cell for some of the mouse clusters is due to a higher rate of multiplexing, leading to lower sequencing depth, for one batch of cells (*Figure 1—figure supplement 1B*). This lower read depth, however, does not result in lower number of genes detected in these mouse cells (*Figure 1—figure supplement 1C*). Cells that met any one of the following criteria were removed from the dataset: <100,000 total reads, <1000 detected genes (counts per million > 0), <75% of reads aligned to genome, CG dinucleotide odds ratio > 0.5, or doublet score > 0.25.

Cells that passed quality control criteria were included in clustering analysis, which was performed using Seurat (*Butler et al., 2018*; *Stuart et al., 2019*). We first performed principal component analysis on the data matrix to construct a KNN graph using the FindNeighbors function in Seurat with k. param set to 20 and using the first 30 PCs. We then clustered the data using FindClusters in Seurat with the resolution parameter set to 0.2 for macaque and 0.4 for human and mouse datasets. The fastMNN implementation of the MNN method was used to correct for donor effects observed in the macaque dataset. Both CCA and fastMNN attempt PCA subspace alignment. The PCA axes with the highest variance can be lost when using CCA. In cases where cell types in different batches are extremely imbalanced, as is the case for macaque where neurons from pulvinar were collected from one donor, CCA might lead to incorrect alignment. The MNN procedure uses a different approach. It finds the nearest neighbors across batches. The difference between the paired cells is then used to infer the magnitude and direction of the batch effect across all PCA subspaces to correct the data. Clustering results from individual species were integrated by employing CCA in Seurat v3. The data was clustered using the first 10 PCs, k.param set to 8, and resolution set to 0.4. To determine robustness of cluster membership, the data was clustered a 100x using 80% of the cells . For every cell, the confidence score is calculated as the number of time it was classified to a particular cluster (*Tasic et al., 2016*).

Differential expression between clusters was calculated with the R package limma using default settings and log2(CPM + 1) expression or using the Seurat 'FindAllMarkers' function. Significantly differentially expressed genes were defined as having greater than twofold change and a Benjamini–Hochberg corrected p-value<0.05 (p<0.01 for donor comparisons). Gene expression distributions of nuclei or cells within a cluster were visualized using dot plots, where the color intensity of the dots represents the average expression level and the size of the dot represents the proportion of cells expressing the gene. Gene Ontology enrichment analysis was performed using ToppGene (https://toppgene.cchmc.org/enrichment.jsp).

## RNA ISH

Single-molecule RNA ISH by RNAscope (Advanced Cell Diagnostics, Newark, CA) was performed as previously described (*Tasic et al., 2018*) using fluorescent kits for mouse tissue and duplex chromogenic kits for human tissue according to the manufacturer's instructions.

## Acknowledgements

We thank the In Vivo Sciences team at the Allen Institute for mouse husbandry, the Tissue Procurement, Tissue Processing, and Facilities teams for assistance with the transport and processing of postmortem and neurosurgical brain specimens, the Molecular Biology department for processing samples for single-cell RNA-sequencing, and the Washington National Primate Research Center for macaque tissue (P51 OD010425). This work was funded by the Allen Institute for Brain Science. We thank the Allen Institute founder, Paul G Allen, for his vision, encouragement, and support.

# Additional information

## Funding
No external funding was received for this work.

## Author contributions
Trygve E Bakken, Conceptualization, Data curation, Formal analysis, Methodology, Writing - original draft, Writing - review and editing; Cindy TJ van Velthoven, Data curation, Formal analysis, Visualization, Writing - original draft, Writing - review and editing; Vilas Menon, Conceptualization, Data curation, Formal analysis, Visualization, Writing - original draft, Writing - review and editing; Rebecca D Hodge, Data curation, Formal analysis, Visualization, Methodology; Zizhen Yao, Data curation, Software, Formal analysis; Thuc Nghi Nguyen, Data curation, Formal analysis, Investigation, Visualization; Lucas T Graybuck, Resources, Data curation, Software; Gregory D Horwitz, Resources, Data curation, Formal analysis, Writing - review and editing; Darren Bertagnolli, Data curation, Formal analysis, Investigation; Jeff Goldy, Data curation, Formal analysis; Anna Marie Yanny, Formal analysis, Investigation, Visualization; Emma Garren, Formal analysis, Investigation; Sheana Parry, Tamara Casper, Soraya I Shehata, Eliza R Barkan, Investigation; Aaron Szafer, Data curation; Boaz P Levi, Kimberly A Smith, Methodology; Nick Dee, Investigation, Methodology; Susan M Sunkin, Project administration; Amy Bernard, John Phillips, Resources; Michael J Hawrylycz, Ed Lein, Supervision; Christof Koch, Supervision, Funding acquisition, Writing - review and editing; Gabe J Murphy, Conceptualization, Data curation, Writing - original draft; Hongkui Zeng, Supervision, Writing - review and editing; Bosiljka Tasic, Conceptualization, Data curation, Formal analysis, Supervision, Methodology, Writing - original draft, Writing - review and editing

## Author ORCIDs
Trygve E Bakken ⓘ http://orcid.org/0000-0003-3373-7386
Cindy TJ van Velthoven ⓘ https://orcid.org/0000-0001-5120-4546
Vilas Menon ⓘ https://orcid.org/0000-0002-4096-8601
Lucas T Graybuck ⓘ http://orcid.org/0000-0002-8814-6818
Gregory D Horwitz ⓘ http://orcid.org/0000-0001-5130-5259
Soraya I Shehata ⓘ https://orcid.org/0000-0002-2909-064X
Amy Bernard ⓘ http://orcid.org/0000-0003-2540-1153
John Phillips ⓘ http://orcid.org/0000-0003-1080-7556
Michael J Hawrylycz ⓘ http://orcid.org/0000-0002-5741-8024
Hongkui Zeng ⓘ http://orcid.org/0000-0002-0326-5878
Bosiljka Tasic ⓘ https://orcid.org/0000-0002-6861-4506

## Ethics
Human subjects: Postmortem adult human brain tissue was collected after obtaining permission from decedent next-of-kin. Postmortem tissue collection was performed in accordance with the provisions of the United States Uniform Anatomical Gift Act of 2006 described in the California Health and Safety Code section 7150 (effective 1/1/2008) and other applicable state and federal laws and regulations. The Western Institutional Review Board reviewed tissue collection processes and determined that they did not constitute human subjects research requiring institutional review board (IRB) review.

Animal experimentation: The brain tissues of two adult Maccaca nemestrina (southern pig-tailed macaque) and one Macaca fascicularis were obtained through the Tissue Distribution Program of the Washington National Primate Research Center and conformed to the guidelines provided by the US National Institutes of Health. All procedures were approved by the Institutional Animal Care and Use Committee of the University of Washington under protocol number 4167-01. All mouse procedures were carried out in accordance with Institutional Animal Care and Use Committee protocols 1508, 1510, and 1511 at the Allen Institute for Brain Science.

Decision letter and Author response
Decision letter https://doi.org/10.7554/eLife.64875.sa1
Author response https://doi.org/10.7554/eLife.64875.sa2

## Additional files

### Supplementary files

• Supplementary file 1. Specimen annotations. All specimens used in this study are listed with associated clustering, donor, and dissection annotations.

• Supplementary file 2. Donor-specific gene signatures in M vs. P neurons. Result of differential gene expression analysis of M vs. P neurons in macaque and human dorsal lateral geniculate nucleus (dLGN) per donor.

• Supplementary file 3. Enrichment of Gene Ontology (GO) terms. Results of GO enrichment analysis of the union of all differentially expressed genes across donors.

• Supplementary file 4. Cluster markers for dorsal lateral geniculate nucleus (dLGN) cell types in macaque, human, and mouse. The table shows the top 20 differentially expressed genes between a given cluster and all other cell types.

• Supplementary file 5. Gene signatures in core vs. shell neurons. Result of differential gene expression analysis of core vs. shell dissected mouse dorsal lateral geniculate nucleus (dLGN) neurons.

• Transparent reporting form

### Data availability

RNA-seq data has been deposited in GEO under accession number GSE182211. Processed count matrices from RNA-seq data are available at ALLEN Brain Map, URL: https://portal.brain-map.org/atlases-and-data/rnaseq/comparative-lgn. Raw sequencing files will be made available at NeMO, URL: https://assets.nemoarchive.org/dat-tfmg0va.

The following datasets were generated:

| Author(s) | Year | Dataset title | Dataset URL | Database and Identifier |
|---|---|---|---|---|
| Bakken TE, van Velthoven CT, Menon V, Tasic B | 2021 | Single-cell RNA-seq uncovers shared and distinct axes of variation in dorsal LGN neurons in mice, non-human primates and humans | https://www.ncbi.nlm.nih.gov/geo/query/acc.cgi?acc=GSE182211 | NCBI Gene Expression Omnibus, GSE182211 |
| Bakken TE, van Velthoven CTJ, Menon V, Tasic B | 2021 | Single-cell RNA-seq uncovers shared and distinct axes of variation in dorsal LGN neurons in mice, non-human primates and humans | https://assets.nemoarchive.org/dat-tfmg0va | NeMo archive, nemo:dat-tfmg0va |
| Bakken TE, van Velthoven CT, Menon V, Tasic B | 2021 | Comparative LGN | https://portal.brain-map.org/atlases-and-data/rnaseq/comparative-lgn | ALLEN Brain Map, comparative-lgn |

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
