## [Decision Letter]

**Acceptance summary:**

This study provides a comparative analysis of the cell variety present in the dorsal lateral geniculate nucleus (dLGN) of mice, non-human primates, and humans using single-cell/single-nucleus RNA-sequencing. The strong and creative bioinformatics analysis used in the study uncovers interesting and subtle cross species links between different types of neurons, providing an extensive characterization of this as yet understudied visual relay nucleus.

**Decision letter after peer review:**

Thank you for submitting your article "Single-cell RNAseq uncovers shared and distinct axes of variation in dLGN neurons in mice, non-human primates and humans" for consideration by *eLife*. Your article has been reviewed by 3 peer reviewers, and the evaluation has been overseen by a Reviewing Editor and K VijayRaghavan as the Senior Editor. The following individuals involved in review of your submission have agreed to reveal their identity: Fenna Krienen (Reviewer #1); Tomomi Shimogori (Reviewer #2); Lucas Cheadle (Reviewer #3).

The reviewers have discussed the reviews with one another and the Reviewing Editor has drafted this decision to help you prepare a revised submission.

Summary:

This manuscript provides a comparative analysis of the cell variety present in the dorsal lateral geniculate nucleus (dLGN) of mice, non-human primates, and humans using single-cell/single-nucleus RNA-sequencing (Smart-seq). The study identifies excitatory and inhibitory dLGN cell types in the three species and shows that the different subclasses of inhibitory neurons are relatively similar across species. In contrast, excitatory neurons appear to bear cross-species differences particularly between mouse and primates.

The study provides an extensive description of the dLGN neurons, an important visual relay nucleus that has been so far poorly studied. As such, these data are very welcomed and will likely attract the interest of researcher working in visual function and beyond. The strong and creative bioinformatics analysis has uncovered interesting and subtle cross species links between different types of neurons. Nevertheless, there a number of aspects that needs to be improved as listed below:

Essential revisions:

1) The introduction will benefit from a clearer explanation of why cross species comparison of dLGN neurons is important. Stating that this comparison has been performed because others have done so for other brain regions is simplistic. Better motivations could be underscored, i.e. species differences in visual system organization such as trichromacy or degree of binocular vision etc. Another motivation could be identifying whether there are discrete vs continuous differences across species related to main cell types. Other can be pointed out.

2) Although the authors have validated a few genes emerged from their analysis (Figures 4G and S4E), the study should include a more extensive and rigorous analysis of cell-type-specific gene expression in all species described either by FISH or ISH. Furthermore, data shown in Figure S3G should be better explained in the figure legend: it is difficult to understand if the cells in question fall into the mentioned magno and parvo cell classes. It may be also useful to compare their data with those of NHP LGN available at https://gene-atlas.brainminds.riken.jp/.

3) The data could be presented more effectively by reorganizing both text and figures with cross-species comparisons from the very beginning. For example, one could start with a figure that has the species-integrated clusters, including only the dLGN dissections, and then explore conservation and divergence of gene expression, proportions etc. In the present form, it is difficult to appreciate the main messages. Each species-specific paragraph contains details that are not directly comparable among species (e.g. connectivity, topography and cell size in humans/NHP, direction sensitivity and dendritic morphology in mouse). What is the main outstanding question: how (or whether) the mouse X, Y and W types map to the M, P and K types? Is the question different?

4) The UMAP projections in Figure 1 need to be labelled for clarity. Colours are not sufficient to interpret the data. This should be also revised for Figure 2-4.

5) The manuscript does not explain how the dLGN shell and the core regions have been dissected. This is important given that these regions are not as clearly distinct as the dLGN lamina in other species. Have the authors taken advantage of the fact that the shell receives input from specific RGCs? Have they used any specific line? Confidence in the dissection could be increased by using a fluorescent approach to selectively label the shell, for example by using a transgenic driver line (Cruz-Martin et al., 2014).

6) The columns in Table S2 need to be labelled to allow interpreting the data and understand statements such as that reported in lines 101 – 103 ("..differentially expressed genes between donors were related to neuronal signalling and connectivity etc..). How have the authors determined that these differentially expressed genes are not related to "activity-dependent effects"?

7) Line 206-207. The data are confusing: are GABA2-6 cell types found in both dLGN and adjacent nuclei in mouse but only in dLGN in primates? The beginning of the paragraph (line 187) seems to suggest that only dLGN datasets were included in the 3 species comparison but it is unclear if this is the case. A clearer cross-species analysis of equivalent regions could further clarify what is conserved in the dLGN proper vs what is shared or distinct in other nuclei.

8) Is PVALB barely expressed in macaque cell types? This is surprising and should be verified. Consistent with the human data (Figure 3), PV protein expression in macaques is detected in some interneurons and in M and P projection neurons (https://pubmed.ncbi.nlm.nih.gov/8885200/). There is a new rhesus macaque genome (Mmul_10) that could help resolving this question.

9) Parameters for clustering analysis (using CCA/Seurat) need to be described, because changes in parameters can change the clusters. Did the authors test if the species integration results hold if parameters are changed? Why GABAergic clusters are different in Figure 4B vs Figure S4A?

10) Figure S1. The data indicate that gene detection is higher for human and mouse than for macaque. However, macaque and human gene detection rates look similar in Figure S1. Can the lower gene detection in macaque be the results of sequencing coverage?

11) Data could be exploited more than what is currently done. For example, the results and discussion mention that DEGs are related to neuronal signalling and connectivity but not metabolic factors. The analysis leading to this conclusion should be shown. Are the two M. nemestrina more similar to each other than they are to the M. fascicularis, or are all 3 donors different from each other in similar ways? Also, is there evidence that there is a fresh vs frozen difference in quality or in the type of genes that are differentially expressed? Similarly it would be useful to describe further how the donor effect magnitudes were compared with previous analysis in Hodge et al. (line 244-245).

12) What do the author mean with "gene expression gradient"? Given that there is a clear anatomical border for each layer in human and monkey LGN, it is difficult to imagine that genes might be expressed in a gradient across layers. Lower magnification ISH image need to be provided to show the existence of a gradient of gene expression.

13) The discussion could be improved by discussing the main conclusions first were discussed first. Is the main point that neuronal types in dLGN that have been defined based on other criteria (morphology, ephys, connectivity) in all species are not very transcriptomically distinct?

---

## [Author Response]

Essential revisions:1) The introduction will benefit from a clearer explanation of why cross species comparison of dLGN neurons is important. Stating that this comparison has been performed because others have done so for other brain regions is simplistic. Better motivations could be underscored, i.e. species differences in visual system organization such as trichromacy or degree of binocular vision etc. Another motivation could be identifying whether there are discrete vs continuous differences across species related to main cell types. Other can be pointed out.

We thank the reviewers for this suggestion. We have revised the introduction extensively to better motivate the evolutionary comparisons between mouse and primate dLGN.

2) Although the authors have validated a few genes emerged from their analysis (Figures 4G and S4E), the study should include a more extensive and rigorous analysis of cell-type-specific gene expression in all species described either by FISH or ISH. Furthermore, data shown in Figure S3G should be better explained in the figure legend: it is difficult to understand if the cells in question fall into the mentioned magno and parvo cell classes. It may be also useful to compare their data with those of NHP LGN available at https://gene-atlas.brainminds.riken.jp/.

Based on the reviewers comment we have adjusted the legend for Figure 3—figure supplement 2E (previously Figure S3G) to improve the description of the marker genes used for ISH on human dLGN sections. Following the reviewer’s suggestion, we have compared the key markers identified in this study to expression in the marmoset atlas. Unfortunately, key markers, like BTNL9 or SUSD2, are either not present in the atlas or show no positive signal. We have now included more extensive ISH analysis of cell types in macaque and human dLGN. The snRNA-seq data identified two different K types in human and macaque dLGN. ISH analysis of *PENK* showed that at least one of the K types is not distributed evenly across the K layers (Figure 3—figure supplement 1D). We also confirmed the presence of GABAergic *SOX14*^+^ and *SOX14*^‑^ cell population in human and macaque dLGN (Figure 5—figure supplement 1E,F). Though we have not quantified the positive cells in the images taken, they do confirm the expansion of the SOX14^-^ population in human dLGN as compared to macaque dLGN which is in line with previously published results (Jager et al., 2021).

3) The data could be presented more effectively by reorganizing both text and figures with cross-species comparisons from the very beginning. For example, one could start with a figure that has the species-integrated clusters, including only the dLGN dissections, and then explore conservation and divergence of gene expression, proportions etc. In the present form, it is difficult to appreciate the main messages. Each species-specific paragraph contains details that are not directly comparable among species (e.g. connectivity, topography and cell size in humans/NHP, direction sensitivity and dendritic morphology in mouse). What is the main outstanding question: how (or whether) the mouse X, Y and W types map to the M, P and K types? Is the question different?

We have updated the introduction to explain the order that we present the results and to motivate the cross-species comparisons. While the morphoelectric properties and connectivity of dLGN neurons have been extensively studied, less is known about their transcriptomic profiles. Therefore, we first characterize the transcriptomic diversity of cells for each species and relate those cell types to previously described cell populations. Then, we integrate data across species to determine the degree of conservation of cell types and gene expression gradients and primate specializations associated with their advanced vision.

4) The UMAP projections in Figure 1 need to be labelled for clarity. Colours are not sufficient to interpret the data. This should be also revised for Figure 2-4.

The schematic in Figure 1B outlines the experimental and data analysis procedures. The UMAP representations shown in panel 1B do not represent original data. To make highlight which datasets are used for integration and that the figure is a schematic representation, we have adjusted the UMAP in Figure 1B. We have added cluster labels to the UMAP representations in Figure 2-4 to make the plot easier to interpret.

5) The manuscript does not explain how the dLGN shell and the core regions have been dissected. This is important given that these regions are not as clearly distinct as the dLGN lamina in other species. Have the authors taken advantage of the fact that the shell receives input from specific RGCs? Have they used any specific line? Confidence in the dissection could be increased by using a fluorescent approach to selectively label the shell, for example by using a transgenic driver line (Cruz-Martin et al., 2014).

The core and shell regions from mouse dLGN have been dissected as follows; Block-face images were captured during slicing at 250 µm intervals. Slices were transferred into dissection dishes containing chilled, oxygenated ACSF.I. Brightfield and fluorescent images of each slice before and after ROI dissection were taken from the dissecting scope. To guide anatomical targeting of core and/or shell enriched dissections, boundaries were identified by trained anatomists, comparing the blockface image and the slice image to a matched plane of the Allen Mouse Brain Common Coordinate Framework version 3 (CCFv3) ontology (Wang et al., 2020). We have added this information to the Methods section of the manuscript and included an example of core and shell dissections in Author response image 1. We agree with the reviewer that using specific transgenic driver lines and/or retrograde tracing strategies to selectively label core or shell neurons could increase dissection confidence. However, the current approach still enables analysis of differential expression between putative core and shell neurons, albeit with reduced power to detect subtle differences, and we validate some of these genes using ISH. We agree with the authors that it will be important to further characterize the neurons that express core- and shell-enriched genes, including their dendritic morphology, electrical properties, and long-range connectivity.

**Author response image 1. sa2fig1:** Example image of core and shell enriched dLGN dissections.

6) The columns in Table S2 need to be labelled to allow interpreting the data and understand statements such as that reported in lines 101 – 103 ("..differentially expressed genes between donors were related to neuronal signalling and connectivity etc..). How have the authors determined that these differentially expressed genes are not related to "activity-dependent effects"?

In response to this comment and comment #11 we have adjusted Supplementary File 2, updated the analysis of donor-specific effects among glutamatergic neurons in human and macaque, and labelled the columns better to allow interpretation of the data. Furthermore, we have included a new Supplementary File 3 that displays the results of gene ontology enrichment analysis of the union of all donor DE genes. This analysis showed significant enrichment for mainly synaptic and axon projection terms. We compared 2495 differentially expressed genes between macaque donors in excitatory neurons to genes up- or down-regulated in dLGN excitatory neurons 8 hours after light exposure of dark-reared mice (Cheadle et al. 2018, Neuron). We found 33 up-regulated genes and 28 down-regulated genes overlapped, and this overlap was not statistically significant based on a hypergeometric test.

7) Line 206-207. The data are confusing: are GABA2-6 cell types found in both dLGN and adjacent nuclei in mouse but only in dLGN in primates? The beginning of the paragraph (line 187) seems to suggest that only dLGN datasets were included in the 3 species comparison but it is unclear if this is the case. A clearer cross-species analysis of equivalent regions could further clarify what is conserved in the dLGN proper vs what is shared or distinct in other nuclei.

We have clarified in the results that all neuronal data were integrated for cross-species comparisons. While we did not sample LGv in primates, we wanted to compare GABAergic neurons from these regions because it has been reported that these cells are distributed differently in the thalamus in primates versus mouse. Indeed, we found that Sox14-negative types that we identified in primate dLGN were not present in mouse dLGN but were found in adjacent nuclei. We also included pulvinar and LP relay neurons as a positive control for the integration since these regions have previously been identified as homologous.

8) Is PVALB barely expressed in macaque cell types? This is surprising and should be verified. Consistent with the human data (Figure 3), PV protein expression in macaques is detected in some interneurons and in M and P projection neurons (https://pubmed.ncbi.nlm.nih.gov/8885200/). There is a new rhesus macaque genome (Mmul_10) that could help resolving this question.

PVALB is expressed in macaque cell types. We thank the reviewer for asking this question as it led us to evaluate the violin plot, that was included in the previous version of the manuscript, was compressed due to a few cells with much higher PVALB expression. We have now changed the expression plots to gene expression dot plots displaying the median expression and proportion of cells expressing the gene. It must be noted that we do expect less robust detection of PVALB in the single nucleus RNA-seq data sets as the majority of PVALB transcripts are located in the cytosol (Bakken et al., 2018).

9) Parameters for clustering analysis (using CCA/Seurat) need to be described, because changes in parameters can change the clusters. Did the authors test if the species integration results hold if parameters are changed? Why GABAergic clusters are different in Figure 4B vs Figure S4A?

We have tested various integration parameters, the integration yielded similar cross-species mapping over a range of variable genes (1000-8000) and PCs (4-40) used for integration. In cases where integration starts to fail the species tend to overlap less (Author response image 2). We have not seen a case where the species integrated well, but the cell types mapped to different cell types than what we reported in the manuscript. We have added the parameters used for clustering to the Methods section of the manuscript.

**Author response image 2. sa2fig2:** Example of the effect of varying cluster parameters on integration of the different species.

10) Figure S1. The data indicate that gene detection is higher for human and mouse than for macaque. However, macaque and human gene detection rates look similar in Figure S1. Can the lower gene detection in macaque be the results of sequencing coverage?

We thank the reviewer for pointing out this discrepancy between the numbers stated in the text and the gene detection rates displayed in Figure 1—figure supplement 1C. In the initial calculations of the gene detection rates, the low-quality cells in the macaque dataset were included whereas the non-neuronal populations in the human dataset were excluded. We have recalculated the gene detection rates. In macaque nuclei the median number of genes detected is ~6,000, and in human nuclei the median number of genes detected is very comparable at ~6,200 genes.

11) Data could be exploited more than what is currently done. For example, the results and discussion mention that DEGs are related to neuronal signalling and connectivity but not metabolic factors. The analysis leading to this conclusion should be shown. Are the two M. nemestrina more similar to each other than they are to the M. fascicularis, or are all 3 donors different from each other in similar ways? Also, is there evidence that there is a fresh vs frozen difference in quality or in the type of genes that are differentially expressed? Similarly it would be useful to describe further how the donor effect magnitudes were compared with previous analysis in Hodge et al. (line 244-245).

We updated the analysis of donor-specific effects among glutamatergic neurons in human and macaque. We included two new SI Tables with the list of differentially expressed genes and enriched gene ontology categories and describe these findings in more detail in the results. For macaque, we compared gene expression of relay neurons (magno- and parvocellular) between each pair of 3 donors (2 females, 1 male) and identified >1000 significantly differentially expressed (>2-fold change; FDR < 0.01) genes. 34-46% of DE genes were shared between pairs of donors. As expected, the two male donors had fewer DE genes (n = 1026) than either male donor versus the female donor (n = 1302 and 1438 genes). The two *M. nemestrina* donors were female, and the *M. fascicularis* donor was male, and it is unclear which genes are differentially expressed due to sex versus species differences. A gene ontology enrichment analysis of the union of all donor DE genes showed significant enrichment for mainly synaptic and axon projection terms (see Supplementary File 3).

We repeated this analysis for each pair of 3 human donors (1 female, 2 male) and found 22-37% of DE genes were shared between donors. The two male donors had fewer DE genes (877) than either male donor versus the female donor (1192 and 1324 genes). In contrast to macaque, a GO analysis of all 2436 LGN DE genes across human donors revealed significantly enriched gene categories related to ribosomal processing rather than neuronal function. This suggests that LGN tissue quality is driving some of the expression differences between human donors and may explain why magno and parvo cell populations were difficult to discriminate in human.

130 of 695 genes that are differentially expressed between neurosurgical and post-mortem donor tissue in middle temporal gyrus (Hodge et al. 2019, ED Figure 2 and SI Table 1) are also differentially expressed between donors in LGN. These genes include many ribosomal genes up-regulated in post-mortem tissue. In Hodge et al., cortical tissue was collected from the same post-mortem human donors that were used in this study. Therefore, we directly compared the magnitude of differential expression in cortical excitatory neurons compared to LGN excitatory neurons using an MTG cluster with similar sampling to LGN (comparable results were seen for other clusters). 457 genes were significantly DE between donors among L2-enriched pyramidal neurons (LAMP5 LTK cluster, n = 809 nuclei), less than 20% of the differences seen in LGN magno/parvo neurons. Many factors could contribute to regional differences in donor effects, including sensitivity to tissue processing, dynamic ranges of cell states, or cell type-dependent effects of genetic background and environment.

12) What do the author mean with "gene expression gradient"? Given that there is a clear anatomical border for each layer in human and monkey LGN, it is difficult to imagine that genes might be expressed in a gradient across layers. Lower magnification ISH image need to be provided to show the existence of a gradient of gene expression.

We agree with the reviewers that this was an unexpected result given the clear anatomical borders between layers. We have restated “gene expression gradient” as a “continuum” of nuclei based on global gene expression profiles from magno and parvo dissections. This continuum was seen in human and not macaque, likely due to lower tissue quality as described in our new analysis of donor expression signatures. A few MC/PC marker genes, such as FOXP2, showed binary expression differences between magno and parvo cells. Other marker genes showed more variable expression, such as the 2 genes in Author response image 3 that are magnocellular-enriched in human (CRH and SUSD2). Both are highly expressed in magno cells, have lower expression in the “middle” parvocellular layers and then have higher expression in the “outer” parvo layers. EPHA7 (parvo-enriched) is shown for comparison and is off in magno and consistently expressed across parvo layers.

**Author response image 3. sa2fig3:** Example ISH panels showing the magnocellular-enriched genes CRH and SUSD2 with lower, but present expression in the parvocellular layer.

13) The discussion could be improved by discussing the main conclusions first were discussed first. Is the main point that neuronal types in dLGN that have been defined based on other criteria (morphology, ephys, connectivity) in all species are not very transcriptomically distinct?

In response to the reviewer’s comments, we have revised the discussion to better highlight the major findings of the paper. This includes highlighting that the transcriptional heterogeneity of thalamocortical neurons is more discrete in primates and more graded in mouse, mouse dLGN core neurons (putative X/Y) align with both M and P neurons and shell neurons (putative W) align with K neurons, and highlighting that the two broad classes of GABAergic types identified here are conserved, and proportions and spatial distributions have shifted dramatically in mouse, non-human primate, and human.

References:

Bakken, T.E., Hodge, R.D., Miller, J.A., Yao, Z., Nguyen, T.N., Aevermann, B., Barkan, E., Bertagnolli, D., Casper, T., Dee, N., et al. (2018). Single-nucleus and single-cell transcriptomes compared in matched cortical cell types. PLoS One 13, e0209648.

Cheadle, L., Tzeng, C.P., Kalish, B.T., Harmin, D.A., Rivera, S., Ling, E., Nagy, M.A., Hrvatin, S., Hu, L., Stroud, H., et al. (2018). Visual Experience-Dependent Expression of Fn14 Is Required for Retinogeniculate Refinement. Neuron 99, 525-539 e510.

Jager, P., Moore, G., Calpin, P., Durmishi, X., Salgarella, I., Menage, L., Kita, Y., Wang, Y., Kim, D.W., Blackshaw, S., et al. (2021). Dual midbrain and forebrain origins of thalamic inhibitory interneurons. *eLife 10*.